# NoxO1 Determines the Level of ROS Formation by the Nox1-Centered NADPH Oxidase

**DOI:** 10.3390/antiox13091113

**Published:** 2024-09-14

**Authors:** Dana Maureen Hebchen, Manuela Spaeth, Niklas Müller, Katrin Schröder

**Affiliations:** 1Institute for Cardiovascular Physiology, Goethe University Frankfurt, 60298 Frankfurt, Germany; 2German Center of Cardiovascular Research (DZHK), Partner Site RheinMain, 60596 Frankfurt, Germany

**Keywords:** NoxO1, Nox1, NoxA1, reactive oxygen species

## Abstract

The Nox1-centered NADPH oxidase complex facilitates the transfer of electrons from intracellular NADPH across the cell membrane to extracellular molecular oxygen, resulting in the formation of superoxide. The complex is comprised of two membrane-bound subunits, namely Nox1 and p22phox, and the cytosolic subunits, namely NoxA1 and NoxO1. The presence of NoxO1 facilitates the proximity of all components, thereby enabling the complex to exhibit constitutive activity. Despite the theoretical sufficiency of all subunits in a 1:1 ratio, the precise composition of the Nox1-centered NADPH oxidase remains unknown. Analyses of mRNA expression in different cell lines revealed an unequal expression of the components, with an excess of NoxO1. Furthermore, plasmid-based overexpression of individual components of the Nox1-centered NADPH oxidase resulted in an excess of NoxO1 mRNA. The objective of this study was to analyze the ability of NoxO1 to control the level of ROS formation by the Nox1 complex. To this end, we generated Hek293 cells for constitutive expression of Nox1 and NoxA1, which were then transfected with increasing concentrations of NoxO1. The data presented herein suggests that ROS formation by the Nox1-centered NADPH oxidase is dependent on the concentration of NoxO1. A surplus of NoxO1 has been observed to exert control over the activity of the complex in accordance with a dose-dependent mechanism. We thus conclude that the ratio of Nox1, NoxA1, and NoxO1 complexes does not adhere to a 1:1 ratio. Conversely, the availability of NoxO1 serves to regulate the formation of ROS by the Nox1-centered NADPH oxidase.

## 1. Introduction

NADPH oxidases are complexes comprising a large membrane-bound subunit (Nox1-Nox5, Duox1, or Duox2) and are responsible for the generation of reactive oxygen species (ROS). It has been demonstrated that Nox1-3 form complexes with the small membrane-bound subunit p22phox and cytosolic subunits that possess either activating (p67phox and NoxA1) or organizing (p47phox and NoxO1) characteristics [1]. To activate Nox1-dependent ROS generation, the cytosolic subunits NoxA1 and NoxO1 must form a complex with the membrane-bound subunits Nox1 and p22phox [2]. NoxO1 mediates constitutive superoxide formation by the Nox1-centered NADPH oxidase [3]. Notwithstanding the prospective formation of an active complex in a 1:1:1:1 stoichiometry of the subunits, a number of homologue-specific mechanisms control the assembly of NADPH oxidase complexes. These mechanisms encompass the involvement of free fatty acids, intracellular trafficking, and post-translational modifications, including phosphorylation, acetylation, and sumoylation [4]. Distinct domains within the subunits facilitate their ability to interact with other members of specific NADPH oxidase complexes. In the case of NoxO1, the presence of a PX (phox homology) domain enables the protein to be translocated to the plasma membrane. A proline-rich region that interacts with the NoxA1 and 2 SH3 (Src homology) domains interacts with p22phox [5].

The formation of reactive oxygen species (ROS) by Nox1, NoxA1, and NoxO1 can result in alterations to a cellular phenotype. It is possible that NoxO1 exerts pro-survival effects in cancer cells. A recently identified biflavonoid has been demonstrated to exert cytotoxic effects on MCF7 breast cancer cells and others through the downregulation of NoxO1 mRNA expression [6]. In PC3 cells, the association of NoxO1 and the deubiquitinase cylindromatosis (CYLD) has been observed to reduce the protein half-life of NoxO1, which subsequently results in a reduction in ROS formation. The knockout of CYLD, and thus the increased abundance of NoxO1, has been observed to promote proliferation, migration, colony formation, and invasion in vitro and in vivo [7]. Additionally, evidence suggests that NoxO1 plays a role in tumor formation in the stomach. In inflammatory settings, TNFα induces NoxO1 expression and subsequent Nox1-dependent ROS formation, which results in the enhanced proliferation of stomach epithelial cells and eventually cancer formation [8]. A potential role for Nox1-centered NADPH oxidase in gastric inflammation has been proposed [9], and there is evidence that inflammatory bowel disease is associated with increased expression of NoxO1, induced by TNFα/NFκB, resulting in higher levels of ROS formation derived from the Nox1-centered NADPH oxidase [10]. Inversely, the deletion of NoxO1 in mice has been observed to result in a reduction in the differentiation of enterocytes and an increased susceptibility to the development of DSS/AOM-induced colitis and cancer formation [11]. Consequently, the potential function of NoxO1 in cancer remains uncertain, underscoring the necessity for a more comprehensive understanding of this NADPH oxidase.

A substantial number of studies have focused exclusively on the expression of Nox1 in the context of NADPH oxidase complex-mediated effects. The objective of this study was to contribute to the comprehension of the function of NoxO1 in the generation of ROS by the Nox1-centered NADPH oxidase.

## 2. Materials and Methods

If not stated otherwise, human genes and proteins are addressed.

### 2.1. Cell Lines and Cell Culture

Human cell lines (Hek293, CaCo2, MDA-MB231; MCF7) were purchased from ATTC (Manassas, VA, USA). All cells were cultured in Minimal Essential Medium (MEM,#11095080, Gibco, Waltham, MA, USA) with 1 mM sodium pyruvate (#M7145, Sigma, St. Louis, MO, USA), 0.1 mM Non-essential Amino acids (#S8636, Sigma), 0.5% Penicillin-Streptomycin (#15140-122, 10,000 U/mL), and 8–20% fetal calf serum (#f7524, Sigma). For MDA-MB231 and MCF7 cells, 0.01 mg/mL human insulin (#I9278, Sigma) was supplemented. Cells were cultured under 5% carbon dioxide atmosphere at 37 °C. Cells were passaged weekly and supplied with fresh culture medium every 3–4 days.

### 2.2. mRNA Expression in Murine Tissue and in Cell Lines

Mice used for the present studies were bred as approved by the authorities. Breeding and animal studies were applied for and approved by the Regierungspräsidium Darmstadt (FU_1223 and FU_1214). In the present study, the mice were not subjected to a treatment but euthanized under anesthesia for organ removal, which according to the German Animal Protection Law is not subject to approval (purpose of organ removal according to §4). For determination of Nox mRNA expression in murine tissue, adult male or female C57Bl6 mice were terminally anaesthetized with isoflurane (Piramal, Mumbai, Maharashtra, India) and subjected to cervical dislocation. Biopsies were taken from the colon, lung, mammary gland, testes, and pancreas. The tissue samples were snap-frozen in liquid nitrogen and stored at −80 °C. Before RNA isolation, tissue was lysed in a Tissue Lyser (Qiagen, Hilden, Germany).

Whole-cell RNA from tissue and cell lines was isolated with the Bio&Sell RNA Mini Kit according to the manufacturer’s instructions. Reverse transcription was performed by rt-SuperScript™ III (#18080093, Invitrogen, Waltham, MA USA) and random primers (#C1181, Promega, Madison, WI, USA). cDNA was amplified in using qRT-PCR in an Aria MX thermocycler (Agilent, Waldbronn, Germany) using ITaq UNIVERSYBR^®^ Green SMX 5000 (#1725125, BioRad, Feldkirchen, Germany) referring to ROX dye as a reference. qRT-PCR samples were performed in technical duplicates. Target gene expression was calculated as relative expression using the ΔΔct method. The primers used in the qRT-PCR are listed in Table 1.

### 2.3. Overexpression Systems

For transient overexpression, cells were seeded on 6-well plates to reach 70–80% confluence and transfected with plasmids coding for Nox1, NoxA1, or NoxO1. All vectors were designed on a pCMV6-Entry backbone with C-terminal myc- and flag-tag for easier detection. Transfection was carried out with 1 µg/mL polyethyleneimine (PEI, #408727, Sigma) or with a Lipofectamine3000^®^ Kit (#L3000001, Invitrogen) for 4–6 h at 37 °C in MEM without supplements. After exchange of MEM to growth media (see Section 2.1), overexpression was allowed for 1 day before performing experiments.

Constitutive overexpression of NoxO1 or NoxA1 and Nox1 was generated by lentiviral transduction followed by selection with 400 µg/mL Hygromycin (#ALX-380-309-G001, Enzo, Lörrach, Germany) or 2 µg/mL Puromycin (#0240.4, Carl Roth, Karlsruhe, Germany). Lentiviral vectors were based on a pLV-EF1a-IRES-Hygro (#85134, Addgene, Watertown, MA, USA) or pHAGE2-EF1aFull-hOct4-F2A-hKlf4-IRES-hSox2-P2A-hcMyc-W-loxP (kindly provided by Dr. Gustavo Mostoslavsky) backbone. Lentiviral particles were produced in Lenti-X™ 293T cells (purchased from Takara, Saint-Germain-en-Laye, France) by transfection with 1 µg/mL PEI together with the packaging plasmids psPAX2/pmD2.G (#12260, #12259, Addgene) and the Nox-encoding plasmid. After 1–2 days, lentiviral particles were harvested from the supernatant and tested with Lenti-X™ GoStix™ Plus (#631280, Takara). Host cells were infected with 1 mL supernatant and 8 µg/mL Polybrene (#TR-1003-G, Merck, Darmstadt, Germany) for 1 day. Selection was started after 1–2 days. As control, cells were transduced with an empty vector construct. Functionality of the overexpressed protein was verified by the ability to produce reactive oxygen species (ROS) after co-transfection of the missing Nox subunits (see Section 2.4). All plasmids were verified by Sanger sequencing at Microsynth Seqlab GmbH (Göttingen, Germany). Table 2 shows an overview of the plasmid constructs used for this study.

### 2.4. ROS Measurement with Chemiluminescence

For ROS measurements, cells were seeded on 6-well-plates to reach 70–80% confluence and transfected with plasmids coding for Nox1, NoxA1, or NoxO1 (see Section 2.3). For the control, cells were transfected with a plasmid coding for eGFP (pEGFP-C1, #2487, Addgene). All ROS measurements reflect spontaneous ROS formation with or without overexpression of the NADPH oxidase subunits indicated.

Living cells were scraped from the dish and resuspended in HEPES-Tyrode buffer (137 mM NaCl #31434-5KG-R, 2.7 mM KCl # P9333, 0.5 mM MgCl # M8266, 1.8 mM CaCl2 # C7902, 5 mM D-Glucose #16301, 0.36 mM NaH2PO4*H2O # 106346, and 10 mM HEPES # H-3375, all from Sigma) with 200 µM L-012 (8-Amino-5-chloro-2,3-dihydro-7-phenyl-pyrido [3,4-d]pyridazine sodium salt, #120-04891, WAKO Chemicals, Neuss, Germany). ROS production was assessed by L-012 chemiluminescence at 37 °C in a 6-channel luminometer. For quenching of the superoxide, 20 U superoxide dismutase (SOD, #S7571, Sigma) was added at the end of each measurement.

### 2.5. Immunofluorescence and Confocal Microscopy

Cells were seeded on 8-well µ-slides (ibidi) and cultivated until 70–80% confluence. In order to determine NoxO1 expression in transient overexpression, cells were transfected with varying amounts of plasmid (see Section 2.3). After 1 day of overexpression, cells were fixed with Roti^®^ Histofix (#P087.1, Carl Roth) for 20 min and washed with Dulbecco’s Phosphate-buffered saline (DBPS, #14040133, Gibco) and 2% L-glycine (#A1377,5000 AppliChem, Darmstadt, Germany) to remove residual paraformaldehyde. Cells were permeabilized with 0.05% Triton-X 100 (#3051.3, Carl Roth) for 10 min. Unspecific binding sites were blocked with 3% bovine serum albumin (BSA, #A8412, Sigma) for 1 h. Myc-tagged NoxO1 was detected by a primary goat-anti myc antibody (#A190-104A, Bethyl/Biomol), diluted 1:300, overnight at 4 °C. The next day, the unbound antibody was removed with 0.03% Tween20^®^ (#12377433, Fisher, Hampton, NH, USA). For 1–2 h, the samples were incubated at room temperature with AF488-conjugated donkey anti-goat secondary antibody (#A11055, Invitrogen), diluted 1:500. Nuclei were stained with 0.1 µg/mL DAPI (4′,6-Diamidino-2-phenylindole dihydrochloride, #D9542, Sigma) for 15 min at room temperature. Finally, slides were washed with 0.03% Tween20^®^ and DPBS, respectively. Slides were stored in the dark until detection with a confocal laser scanning microscope (LSM800, Zeiss, Jena, Germany). Images were processed with Zen blue (3.3 edition, Zeiss) and ImageJ (1.53q).

### 2.6. NoxO1 Protein Detection

Hek293 cells transiently overexpressing NoxO1 were lysed (250 mM Tris*HCl pH7.4 #AE15.3 Carl Roth, 750 mM NaCl #S/3160/65 Fisher, 50 mM Na4PPi # 106,391 Merck, 100 mM NaF #201154 Sigma, 10% Triton-X #3051.3 Carl Roth, 2 mM Orthovanadate #A2196 AppliChem, 10 mM Okadaic Acid #ALX-350-011 Enzo, 200 µM PMSF #6367.1 Carl Roth, 20 µM cOmplete ##4693116001, Merck). After centrifugation (10 min, 4 °C, 13,000 rpm), the nuclear fraction was discarded. Total protein amount in the cytosolic fraction was quantified by a photometric Bradford assay with Roti-Quant^®^ (#K015.1, Carl Roth). The samples were boiled for 10 min at 95 °C in Laemmli buffer (25.5% glycerol #G7893 Sigma, 6% SDS # L-4390 Sigma, 188 mM Tris*HCl pH 6.8 #AE15.3 Carl Roth, 60 mM DTT # A1101 AppliChem, 0.04% bromphenolblue #A3640,0005 AppliChem). Proteins were separated by Sodium-Dodecylsulfate-Polyacrylamide-Gel-Electrophoresis (SDS-PAGE) on 10% acrylamide gels followed by a Western blot. Overexpressed NoxO1 was detected by an antibody sold by Abcam (Cambridge, UK) (#AB97788) and through flag- or myc-tag (#A190-104A Bethyl, #F7425 Sigma) and normalized to Erk1/2 (#4696, Cell Signaling, Leiden, The Netherlands) expression.

### 2.7. Statistical Analysis

Data are presented as mean and standard error of the mean (SEM). All experiments were at least conducted in three independent biological samples with duplicates. Each sample is defined by “n”. Calculations and statistical analysis were performed with Prism 9 (Graph Pad, prism) and Excel2016.

## 3. Results

### 3.1. NADPH Oxidase Subunits Are Differentially Expressed in Cancer Cells

The literature reports disparate findings regarding the tissue-specific expression of NoxO1 across distinct databases. ProteomicDB [12] indicates that the NoxO1 protein is predominantly expressed in the stomach, colon, and rectum in humans and mice. However, the GTExPortal [13] indicates that NoxO1 mRNA is predominantly expressed in the testis, brain, and small intestine in humans. Our own analysis revealed a high level of NoxO1 mRNA expression in murine colon, breast, and pancreas tissue (see Appendix A). Given the potential role of NoxO1 in cancer, as outlined in the introduction, we conducted an analysis of human colon and breast cancer cell lines (CaCo2, MCF7, and MDA-MB231) in addition to Hek293 cells, which served as an easily manipulatable model cell line. The gene expression data for the various cell lines can be accessed via the Human Protein Atlas [14]. The results of mRNA expression as nTPM (normalized transcripts per million) of the NADPH oxidase subunits as published by the Human Protein Atlas are displayed in Appendix A. The tau specificity score is employed as a numerical indicator of the degree of specificity of gene expression across cells or tissue. The value ranges from 0 to 1, with 0 indicating identical expression across all cells and 1 indicating expression in a single cell type. The table illustrates a multitude of expression levels around zero, while Tau indicates highly specific mRNA expression patterns. It is noteworthy that p22phox and Rac1 exhibit the lowest cell-specific expression and the highest nTPM values. It can be concluded that these two subunits are expressed at high levels in almost every cell or tissue, and thus do not represent potential therapeutic targets. In contrast, the majority of other subunits of the NADPH oxidase may exhibit expression levels that are insufficient for comprehensive analysis in untargeted approaches, such as RNA sequencing. As stated by Zhao et al. (2020), the application of transcriptome profiling based on TPM, as observed in the Human Protein Atlas database, has resulted in the inadvertent misuse of these methods for investigations of differentially expressed genes. The authors highlight that TPM is highly susceptible to sample preparation protocols and should not be employed for the comparison of disparate tissue components or even different regions of the same cell [15]. Consequently, an analysis was conducted utilizing specific primers for the low-expressed NADPH oxidase subunits in Hek293 cells (Appendix A). In contrast to the Human Protein Atlas, our findings revealed higher expression levels of Nox2 mRNA than Nox1 and higher expression levels of the subunits NoxA1 and NoxO1 than p47phox and p67phox. Upon analyzing the expression of the large membrane-bound subunits of each NADPH oxidase in CaCo2, MCF7, MDA-MB231, and Hek293 (Figure 1), it was observed that Nox1 mRNA exhibited the highest expression levels in CaCo2 cells. The highest expression of Nox2 mRNA was observed in Hek293 cells, followed by MDA-MB231 cells.

Nox3 is largely exclusive to the inner ear, and accordingly, if expressed at all, Nox3 mRNA was expressed at very low levels throughout all analyzed cell lines. Nox4 mRNA expression was low, while CaCo2 and MCF7 cells showed the highest values. Nox5 mRNA expression was highest in MCF7, while Duox1 and 2 mRNAs were highest in MDA-MB231 cells.

The relatively low expression of Nox1 is unexpected given the well-described role of Nox1 in cancer. However, as previously stated, Nox1 is incapable of producing ROS in isolation. Accordingly, an analysis was conducted on the expression of its cytosolic subunits, NoxA1 and NoxO1.

### 3.2. NoxO1 Expression Is Often Higher than That of Nox1 and NoxA1 Even in Overexpression Settings

A comprehensive examination of the constituents of the Nox1-centric NADPH oxidases revealed an excess of NoxO1 mRNA in most selected cell lines, except for CaCo2 (Figure 2).

Upon overexpression of Nox1, NoxA1, and NoxO1 (0.15 µg/cm^2^) in the backbone plasmid pCMV.6-entry in Hek293 cells, it was observed that the expression of the components was unequal. It is anticipated that the transfection efficacy of plasmids with an identical backbone in a single cell line will yield comparable expression levels of the transfected genes. In contrast, our data indicate that, at least in the case of Hek293 cells, the overexpression of one subunit induces the expression of other components of the complex (Appendix A).

As qPCR analysis of mRNA is susceptible to error [16] and does not indicate the spatial distribution of the protein overexpressed, we analyzed protein expression in Hek293 cells overexpressing NoxO1 using myc- and flag-tag antibodies (Figure 3).

In immunofluorescence, it was observed that NoxO1 accumulated in the perinuclear endoplasmic reticulum when expressed at plasmid concentrations below 0.1 µg/cm^2^. However, when transfected with 0.1 or 0.2 µg plasmid/cm^2^, NoxO1 was predominantly observed at the plasma membrane. With increasing plasmid load, NoxO1 was found to be distributed throughout the entire cell (Figure 3B). It is noteworthy that the transfection efficacy, and consequently the number of cells expressing NoxO1, demonstrated a relatively consistent trend. In contrast to immunofluorescence, the Western blot analysis detecting the flag-tag of the overexpressed NoxO1 demonstrated robust expression of the NoxO1 protein even when cells were transfected with a minimal amount of plasmid DNA, at 0.01 µg/cm^2^. The expression of NoxO1 was observed to increase in conjunction with an escalation in plasmid concentration, reaching a peak at 0.1 µg/cm^2^. In light of the aforementioned experiments, it can be concluded that a concentration of 0.1 to 0.2 µg/cm^2^ of the NoxO1 plasmid is sufficient to achieve the optimal expression of the protein at the plasma membrane. Western blots for NoxO1 and myc-tag corroborated the findings obtained with the flag-tag antibody (Figure 4).

The overexpression of a tagged NoxO1 cDNA in HEK293 cells has been observed to result in a dose-dependent increase in protein expression of the tagged version, while the endogenous NoxO1 level remains constant. It is noteworthy that the NoxO1 antibody appears to detect specific bands between 110 and 300 kDa. In light of the possibility that SDS and the other sample treatment methods are insufficient to destroy all cellular complexes, it can be postulated that the bands of high molecular weight represent NoxO1 in complexes. It must be acknowledged that at this juncture, any conclusions regarding the composition of these complexes remain purely speculative. Nevertheless, the likelihood of the abundance of NoxO1 in complexes is high.

### 3.3. Overexpressed NoxO1 Is Functional and Dose Dependently Induces ROS Formation by the Nox1-Centered NADPH Oxidase

In order to ascertain the functionality of the produced protein, superoxide formation was measured, given that the sole known function of NoxO1 is to mediate ROS formation by the Nox1-centered NADPH oxidase. The use of L-012 as a reagent for superoxide detection by NADPH oxidases [17] yielded no evidence of basal superoxide formation in the analyzed cells. Transfection of CaCo2 cells with the components of the Nox1-centered NADPH oxidase resulted in the generation of ROS, whereas no such generation was observed in MCF7 and MDA-MB231 cells (Appendix A). The use of an increased plasmid quantity ultimately led to the generation of ROS in MCF7 and MDA-MB231 cells, with a level approximately half that observed in CaCo2 cells transfected with 0.01 µ/cm^2^. This may be attributed to a low transfection or expression efficacy.

Upon overexpressing all components of the Nox1-centered NADPH oxidase with plasmid concentrations of 0.01 µg/cm^2^ each in Hek293 cells, no augmented L-012 signal was discerned in comparison to GFP-transfected cells. Furthermore, the superoxide formation did not exhibit a notable increase when the plasmid concentrations exceeded 0.05 µg/cm^2^ for each plasmid (Appendix A).

Hek293 cells with constitutive NoxO1 expression (Hek293-cNoxO1) were generated, exhibiting a 500-fold increase in NoxO1 mRNA expression compared to native Hek293 cells (Appendix A). The same outcome was observed in these cells as was seen in native Hek293 cells overexpressing all subunits, namely that no ROS formation was observed at 0.01 µg plasmid/cm^2^ and that a plateau was reached at 0.05 µg plasmid/cm^2^ (Appendix A). It was therefore concluded that in triple-transfected cells with more than 0.05 µg/cm^2^ of each plasmid, the individual expression of the subunits may interfere with each other, preventing the further formation of proper Nox1-centered NADPH oxidases. To address this issue, Hek293 cells with constitutive expression of Nox1 and NoxA1 were generated (Hek293-cNox1/NoxA1), which expressed 1200 times more Nox1 and 500 times more NoxA1 mRNA than native Hek293 cells (Appendix A). This is equivalent to transient transfection with plasmid concentrations of 0.015 µg/cm^2^ for Nox1 and NoxA1. The cells were transfected with increasing concentrations of plasmid encoding NoxO1. The overexpression of NoxO1 from 0.01 to 0.15 µg plasmid/cm^2^ resulted in an exponential increase in ROS formation (Figure 5).

Given that the efficacy of transfection remains unaltered by increased plasmid concentrations, as illustrated in Figure 3A, the data indicate that a tenfold increase in NoxO1 plasmid concentration per cell approximately triples ROS formation.

## 4. Discussion

The differential expression patterns of NADPH oxidases observed in cancer cells of various etiologies, as well as in Hek293 cells, shed light on the complexities of ROS regulation in different cellular environments. Despite the widespread belief that cancer cells typically exhibit elevated ROS levels, our findings showed no basal superoxide production in any of the cells analyzed [18]. This unexpected result could be attributed to the specific culture conditions under which the experiments were conducted, highlighting the importance of considering environmental factors when studying ROS production. It is conceivable that freshly isolated cancer cells or tissue samples may present a different profile, potentially revealing basal superoxide generation that is not evident in cell culture systems.

A notable finding of this study was that NoxO1 mRNA is more prevalent than Nox1 and NoxA1 in the majority of cells. Furthermore, when equivalent quantities of plasmids for Nox1, NoxA1, and NoxO1 were overexpressed in Hek293 cells, an unequal expression of the mRNAs was observed. In particular, the expression of NoxO1 was found to exceed that of the other two genes. This suggests a regulatory mechanism favoring NoxO1 expression, which could play a critical role in the formation and function of the Nox1-centered NADPH oxidase complex. In native cells and tissue, NoxA1 expression is frequently less abundant than that of NoxO1 or Nox1 [19]. Hill et al. observed that NoxA1 expression in the placenta is lower than that of NoxO1, and both are lower than Nox1, which appears to be the primary NADPH oxidase in murine placenta. In the fetal brain, Nox1, NoxA1, and NoxO1 are expressed at an equal level. In contrast, the fetal liver exhibits a higher expression of NoxA1 compared to NoxO1, both of which are expressed at a level higher than that of Nox1 [20]. In murine lung tissue, the expression of Nox1, NoxA1, and NoxO1 is nearly equal. In the colon and heart, Nox1 is expressed at a higher level than NoxA1 and NoxO1 [21]. It is therefore evident that the expression of the subunits of the Nox1-centered NADPH oxidase is regulated in a tissue-specific manner. The tissue-specific regulation of these subunits, as highlighted by the varying expression patterns across different organs, underscores the complexity of ROS regulation and the potential for differential physiological and pathological roles of Nox1, NoxA1, and NoxO1. Further research is required to elucidate the mechanisms underlying this regulation and to identify the factors that determine the individual expression levels. In this study, we initiated an investigation into the consequences of unequal expression of the subunits.

The sole known function of the Nox1-centered NADPH oxidase subunits is to form the complex, which is a prerequisite for ROS formation. Once assembled, the complex remains constitutively active until it dissociates. Despite the identification of multiple phosphorylation sites in all three components, NoxO1 and NoxA1 have been shown to enable constitutive superoxide formation by the Nox1 complex. Another constitutively active NADPH oxidase is Nox4, which possesses a distinctive attribute within the family of NADPH oxidases: the direct formation of H_2_O_2_ [22]. Consequently, Nox4 has been demonstrated to prevent DNA damage and inflammation [23]. In contrast, the superoxide-producing Nox1 has been demonstrated to promote DNA damage [24]. The detrimental effects of Nox1 activity may be averted by the knockout of NoxO1, which has been demonstrated to safeguard female mice from atherosclerosis [25] and to promote longevity in mice [26]. Concurrently, mice lacking NoxO1 exhibit augmented angiogenesis [27], which may potentially enhance tumor formation [11].

The example of two different NADPH oxidases sharing the ability of constitutive ROS formation but acting at different sides with either H_2_O_2_ or superoxide demonstrates that the inhibition of electron transport over the membrane within the large membrane-bound subunit is not an optimal strategy for preventing ROS formation by one NADPH oxidase complex without reducing the activity of the other. This approach carries the risk of non-specific inhibition of unintended NADPH oxidases. Our findings suggest that targeting cytosolic subunits could offer a more specific strategy for inhibiting individual NADPH oxidases. A deeper comprehension of the molecular mechanisms that activate the Nox1/NoxA1/NoxO1 complex may facilitate the identification of suitable inhibitors [28].

The principal conclusion of this manuscript is that NoxO1 exerts control over superoxide formation by the Nox1-centered NADPH oxidase. It can therefore be concluded that Nox1 expression is an unreliable indicator of oxidative stress levels. Consequently, it is essential to ascertain the expression levels of all subunits, with particular emphasis on NoxO1. However, the reliance on an overexpression system in this study introduces limitations, as these conditions may not accurately reflect the in vivo environment. Therefore, further research is needed to validate these findings under physiological conditions and to explore the potential for developing selective inhibitors that can modulate ROS production in a controlled and specific manner.

## 5. Conclusions

In conclusion, this study provides new insights into the differential expression and regulation of NADPH oxidase subunits, particularly emphasizing the critical role of NoxO1 in modulating superoxide production by the Nox1-centered NADPH oxidase. Our findings challenge the conventional view that Nox1 expression alone is indicative of oxidative stress, underscoring the necessity of assessing the expression levels of all relevant subunits, especially NoxO1, to obtain a more accurate understanding of ROS regulation in different cellular contexts. The observed tissue-specific expression patterns of Nox1, NoxA1, and NoxO1 further highlight the complexity of ROS regulation and its potential physiological and pathological implications. However, the use of an overexpression system in this study imposes limitations, as it may not fully replicate in vivo conditions. Future research should aim to validate these results under physiological conditions and could offer a more precise approach to modulating ROS production in specific NADPH oxidase complexes.

## Figures and Tables

**Figure 1 antioxidants-13-01113-f001:**
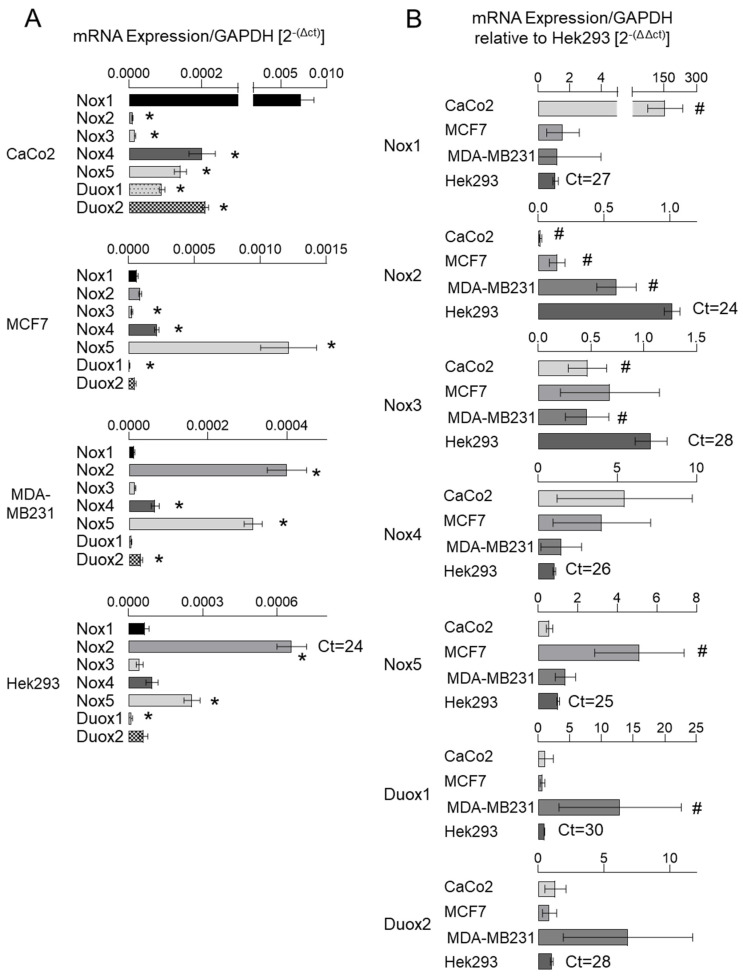
Nox family mRNA expression in cancer cell lines. mRNA expression of individual large membrane-bound subunits depicted as copy per copy of GAPDH (mean ct 15 ± 1) (**A**) and relative to mRNA expression in Hek293 cells (**B**). Mean ± SEM; n = 4; * *p* < 0.05 (gene of interest vs. Nox1); # *p* < 0.05 (mRNA expression in the cell of interest vs. Hek293).

**Figure 2 antioxidants-13-01113-f002:**
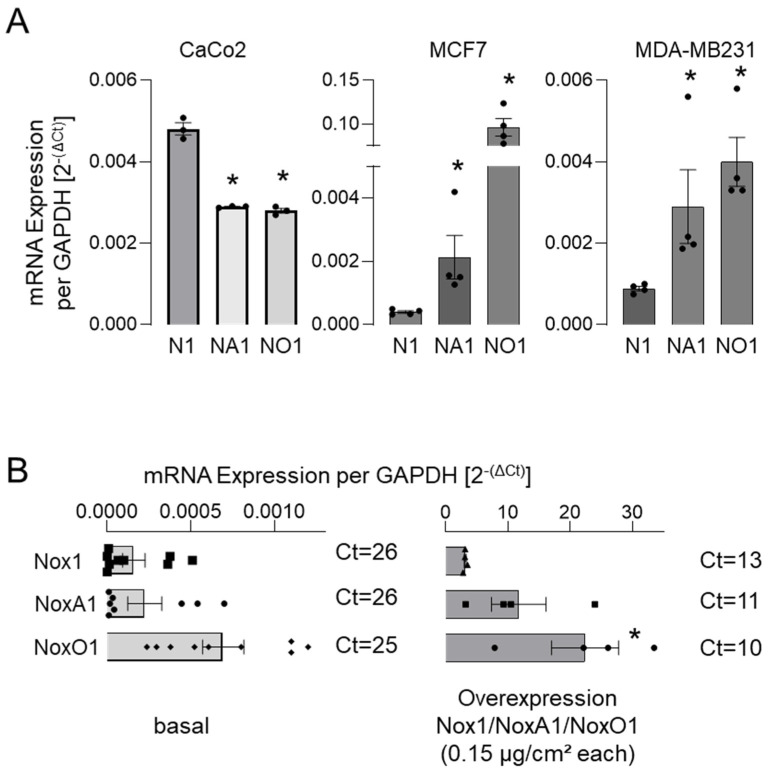
mRNA expression of subunits of the Nox1-centered NADPH oxidase in Hek293. mRNA expression of Nox1 (N1) and of individual cytosolic subunits NoxA1 (NA1) and NoxO1 (NO1) depicted as copy per copy of GAPDH (mean ct 15 ± 1) (**A**) in some cancer cell lines, (**B**) in untreated Hek293 cells, and in Hek293 cells overexpressing Nox1, NoxA1, and NoxO1. Mean ± SEM; n ≥ 4; * *p* < 0.05 (gene of interest vs. Nox1).

**Figure 3 antioxidants-13-01113-f003:**
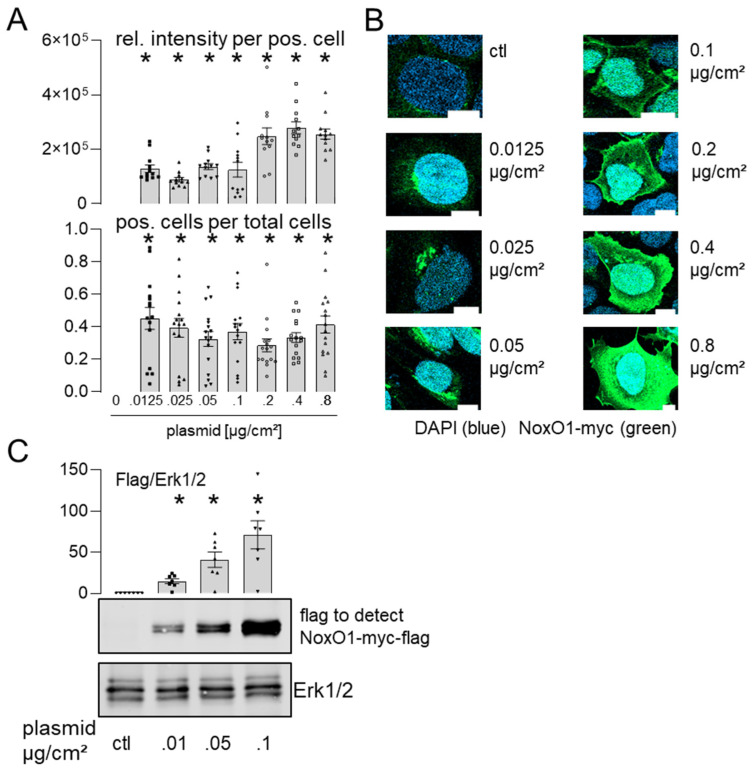
NoxO1 cDNA overexpression results in a dose-dependent expression of NoxO1 protein upon overexpression in Hek293 cells. NoxO1 overexpression in Hek293 cells. (**A**) Statistics of immunofluorescence of relative intensity of NoxO1 staining per positive cell and relative number of transfected cells per well; (**B**) representative immunofluorescence images of cells transfected with increasing plasmid concentrations. (**C**) Western blot for overexpressed NoxO1-myc-flag with statistics for relative expression. Mean ± SEM; n = 15 (3 × 5 cells) or 5 (Western blot), scale bar = 10 µm; * *p* < 0.05 (myc-positive fluorescent immunostaining or protein expression in cells transfected with the indicated amount of plasmid of NoxO1-mcy vs. empty vector).

**Figure 4 antioxidants-13-01113-f004:**
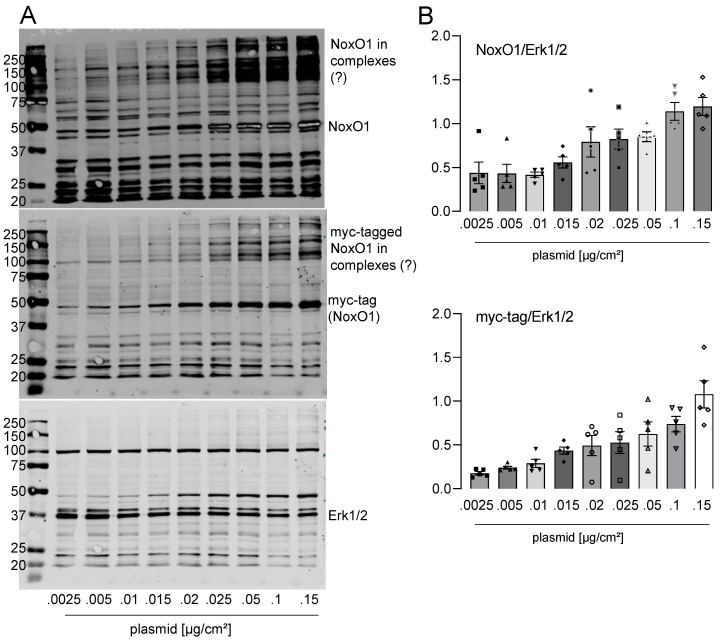
NoxO1 cDNA overexpression in Hek293 cells results in a dose-dependent increase in NoxO1 protein. NoxO1 overexpression in Hek293 cells. (**A**) Full original Western blots with detection of NoxO1 (rabbit), myc-tag (goat) and Erk1/2 (mouse), sequentially on the same blot and (**B**) Statistics of relative intensity of NoxO1 or myc-tag staining per Erk 1/2 staining. Mean ± SEM; n = 5.

**Figure 5 antioxidants-13-01113-f005:**
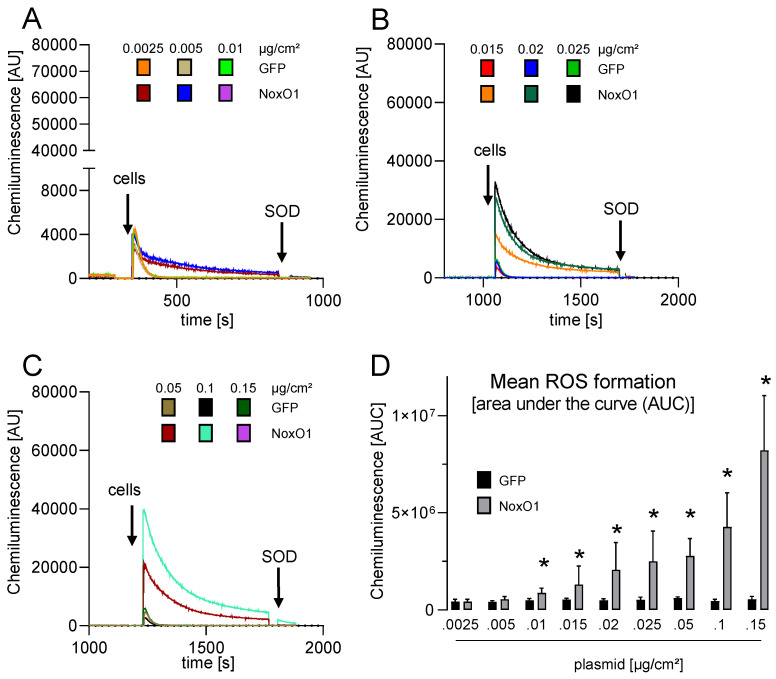
ROS formation in HEK293 with constitutive overexpression of Nox1 and NoxA1 as measured by L-012 mediated chemiluminescence. NoxO1 over expression at accelerating concentration in Hek293-cNox1/NoxA1 cells. (**A**–**C**) original traces and (**D**) statistics of ROS formation in cells transfected with the indicated amount of plasmid, mean ± SEM; n = 3; * *p* < 0.05 NoxO1 vs. GFP.

**Table 1 antioxidants-13-01113-t001:** Primers for qRT-PCR.

mRNA (Accession No.)	Forward (5′-3′)	Reverse (5′-3′)
*GAPDH* *(NM_002046.7)*	TGCACCACCAACTGCTTAGC	GGCATGGACTGTGGTCATGAG
*mNoxo1 (NM_027988.4)*	TGGAGGAGGTAGCAACGTGC	AGAGCGACTGCCCTCGTAGG
*hNox1* *(NM_007052.5)*	TCTTATGTGGCCCTCGGACT	CCAGACTGGAATATCGGTGACA
*hNoxA1* *(NM_006647.2)*	TGGGAGGTGCTACACAATGTG	TTGGACATGGCCTCCCTTAG
*hNoxO1* *(NM_144603.4)*	GAGATCTGACCGCGTTCTCC	CAGCAGCCTCCGAGAATAGG
*p47phox* *(NM_000265.7)*	ACGAGTTCCATAAAATGCTGAAGG	GAGATCTTCACGGGCAGTCC
*p67phox* *(NM_000433.4)*	ACCTTGAACCAGTTGAGTTGCG	GTCGGACTGCGGAGAGCTT

**Table 2 antioxidants-13-01113-t002:** Overview of plasmid constructs used.

Protein Expressed		Backbone	Tag for Detection
Nox1	transient	pCMV.6-entry	c-myc, Flag-DDK
NoxA1			
NoxO1			
eGFP		pEGFP-C1	GFP
(empty vector)	constitutive	pLV-EF1a-IRES-Hygro	-
NoxO1			
Nox1 + NoxA1		EF1aFull-hOct4-F2A-hKlf4-IRES-hSox2-P2A-hcMyc-W-loxP	

## Data Availability

The data presented in this study are available on request from the corresponding author.

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
