# Peer review of "NoxO1 Determines the Level of ROS Formation by the Nox1-Centered NADPH Oxidase"

_antioxidants, 2024, doi:10.3390/antiox13091113_

Round 1

Reviewer 1 Report (Previous Reviewer 1)

In their revised manuscript of Hebchen  et al. adequately addressed all the issues that this reviewer noted in the previous submission. This reviewer has no further comments. 

In their revised manuscript of Hebchen  et al. adequately addressed all the issues that this reviewer noted in the previous submission. This reviewer has no further comments. 

Author Response

In their revised manuscript of Hebchen  et al. adequately addressed all the issues that this reviewer noted in the previous submission. This reviewer has no further comments.

Answer: We thank the reviewer for their time to take a look at our manuscript again.

Reviewer 2 Report (New Reviewer)

The manuscript entitled “NoxO1 determines level of ROS formation by the Nox1 centered NADPH oxidase “reports the mRNA expression of NOXs and Nox1 components in several cancer cell lines under normal and overexpression of Nox1 components conditions, and ROS formation of several cancer cell lines under overexpression conditions of Nox1 components. Overall, the results indicated and discussion are not enough and quantitative to support the conclusion described in the abstract. In the present form, this article is not informative to the reader.

<Major concerning issues>

1. It should be discussed whether the mRNA amounts and protein amounts expressed are correlate or not at low mRNA amount indicated in Figures 1 and 2.

2. In the abstract the authors describe that surplus NoxO1 forces the activity of Nox1.

(1) I think “surplus” is vague: is there any data representing the stoichiometry of Nox1 subunit proteins under the experimental conditions in this manuscript? Is there any evidence supporting that two or more NoxO1 can interact with one Nox1 complex, and the interaction with second NoxO1 enhances the activity of Nox1?

(2) Is it possible to exclude the possibility that an excess NoxO1 modulate the redox homeostasis, which induces ROS production?

3. In the figure panel of Figure 2 for “Overexpression Nox1/NoxA/NoxO1” conditions of “CaCo2” cell there are data points close to 0. Why? Also, in the lower figure panel of Figure 3A, there are data points with near 0 values. What does it indicate?

4. The results in Figure 2 suppose that CMV promoter were not fully functionable in MCF7 and MDA-MB231 cell lines. Are these phenomena specific to Nox1/NoxA1/NoxO1 genes? Is there any paper reporting an expression of any protein utilizing same CMV promoter in the same cancer cell lines?

5. At page 5 line 15 from top, “the potential role of NoxO1 in cancer” is not clear for me: does inflammation induced by ROS/NoxO1 lead the cancer, or is an expression of Nox1/NoxO1 enhanced in cancer cells? The details of “the role” and mechanism should be described with proper references.

<Minors> check and polish the English carefully again. Followings are the examples.

Page 1, line 11 in the abstract: I can‘t understand the phrase “… which than were …”.

Page 9, line 3-4 from bottom: the sentence “Those cells … “seems grammatically incorrect.

Page 11, line 18: what does “electron formation” mean?

Figure 4

(1) The legend is incomplete. Add the explanation of each figure panels.

(2) The areas under the curve at 0.02 and 0.025 μg/cm2 NoxO1 seem larger than that of at 0.05 μg/cm2 NoxO1. Please check the values in Figure 4D again.

Author Response

We prepared a word document for reviewer 2.

Round 2

Reviewer 2 Report (New Reviewer)

The revised version of the manuscript is refined adequately.

Recheck Figure 4A: is the posion of "Erk1/2" correct?

Author Response

The revised version of the manuscript is refined adequately.

Recheck Figure 4A: is the posion of "Erk1/2" correct?

We thank the reviewer for reviewing our manuscript again. We are not sure, what position the reviewer is referring to. We think the position within the figure should be like it is. In case the reviewer is referring to the size of the detected bands on the Western blot in relation to the marker: Yes, this is Erk1/2, which appears around the marker band for 37 kDa. We however do not know, if this is due to the lot of the marker or because it runs differently in the gel we used.

This manuscript is a resubmission of an earlier submission. The following is a list of the peer review reports and author responses from that submission.

Round 1

Reviewer 1 Report

Comments and Suggestions for Authors

The revised manuscript of Hebchen et al. did partially address the concerns of this reviewer however, it still needs minor correction/precisions before it can be accepted for publication. 

This reviewer notes the following issues.

1.     Methods: Page 4, Lines 128-129. Please mention here the reagents, their relative specificity and the applied concentrations (Lucigenin, Amplex-red and Luminol-HRP).

2.     Page 9, Figure 4 Legends: “Statistics (A) of Immunohistochemistry (B) of relative intensity of NoxO1 staining per positive cell and relative number of transfected cells per well” correct to “(A) Statistic of immunofluorescence of relative intensity of NoxO1 staining per positive cell and relative number of transfected cells per well (B)representative immunofluorescence images of cells transfected with increasing plasmid concentrations.”

3.     Page 9, Line 211: “NoxO1 mainly appears at the plasma membrane and with increasing plasmid load the whole cell is flooded with NoxO1 (Figure 4A&B).”, correct to “NoxO1 mainly appears at the plasma membrane and with increasing plasmid load the whole cell is flooded with NoxO1 (Figure 4B). 

4.     Figure 4 Figure Legend part C is still missing

5.     Page 11, Line 252: “No hydrogen specific signal”, correct to “No hydrogen peroxide-specific signal ”

6.     Page 12, Line 266: “An interesting finding of the present manuscript was that NoxO1 mRNA is more abundant than that of Nox1 and NoxO1” , correct to “An interesting finding of the present manuscript was that NoxO1 mRNA is more abundant than that of Nox1 and NoxA1. 

7.     Mistyping:

-       Page 1, Line 17 : “We thought to analyze”, correct to “We sought to analyze”

-       Page 2, Line 54: “inflammatory bowl”, correct to “inflammatory bowel” 

Comments on the Quality of English Language

English language is fine, minor mistypings need to be addressed.

Author Response

We thank the reviewer for careful reading of our manuscript. The material and method section was extended. For 2.-7., we revised the new version of the manuscript accordingly.

Reviewer 2 Report

Comments and Suggestions for Authors

The revised version of the manuscript is substantially very similar to the first version, and I keep having the same concerns.

·         Figure 1 to 3: the qPCR quantifications have been made according to the dCT methods taking GAPDH as an internal control (IC). This is right for comparison of the expression of a certain gene in different samples. However, you cannot compare the Ct value of different genes, even when using the same IC. To compare the expression of Nox1, NoxA1 and NoxO1 absolute quantification is required.

·         Figure 5: In HEK293 cells that overexpress Nox1 and NoxA1, upon transfection with NoxO1 the authors observe an increase in ROS production. So, the authors conclude that a surplus of NoxO1 stabilises the complex. In line with my previous comment, to sustain this notion, evidence regarding the expression ratio of the three components in the different transfection conditions should be provided (1:1:1, 1:1:2, 1:1:3, etc). Otherwise the results do not support the author´s conclusion.

Author Response

  • Figure 1 to 3: the qPCR quantifications have been made according to the dCT methods taking GAPDH as an internal control (IC). This is right for comparison of the expression of a certain gene in different samples. However, you cannot compare the Ct value of different genes, even when using the same IC. To compare the expression of Nox1, NoxA1 and NoxO1 absolute quantification is required.

We thank the reviewer for giving us the opportunity, to make our view more clear.

It is not our intention to compare Nox expression levels over cells. Absolute quantification would be required for certain mRNA in different samples if the goal were to get a glimpse on total number of mRNAs in different samples. In our study, the goal is to get an impression of the intracellular relations of the individual Nox subunits. Even if the efficiency of the individual primers is not the same, the error will be the same in all samples. Accordingly, the delta ct method allows for relative comparison of the expression level of different mRNAs in individual samples.

  • Figure 5: In HEK293 cells that overexpress Nox1 and NoxA1, upon transfection with NoxO1 the authors observe an increase in ROS production. So, the authors conclude that a surplus of NoxO1 stabilizes the complex. In line with my previous comment, to sustain this notion, evidence regarding the expression ratio of the three components in the different transfection conditions should be provided (1:1:1, 1:1:2, 1:1:3, etc). Otherwise, the results do not support the author´s conclusion.

Unfortunately, we disagree with the reviewer. Even if the exact relation of the individual subunits is not known, the conclusion is based on the fact that in cells with stable overexpression of Nox1 and NoxA1, a stepwise increased surplus of NoxO1 results in elevated level of ROS as measured by L-012. If Nox1 centered NADPH oxidase, is the only source of ROS targeted by NoxO1, such an increase indicates better function of the complex. The function of NoxO1 within the Nox1 centered NADPH oxidase is to organize the subunits together, or in other words to stabilize the interaction of the complex components. We kindly ask for the reviewers understanding that we keep the conclusion drawn.

Reviewer 3 Report

Comments and Suggestions for Authors The authors have answered most questions and comments satisfactorily. However, the legend of figure 4C is still missing in the revised manuscript

Author Response

We are very sorry. Obviously, the reviewer saw an older version of the manuscript. The version (.docx) that can be downloaded from the journals server has 4C in it. We hope, the complete new submission will solve the problem.

Round 2

Reviewer 2 Report

Comments and Suggestions for Authors

Dear authors,

The manuscript is basically the same as in the first submission, and I am afraid that we disagree in some key aspects. My opinion regarding the mns has not changed.

So if you need any further explanations, please ask the Editor,

Sincerely

Author Response

We wrote a letter with figures to reply to the editors comment.
